# Implications of Climate Change on Wind Energy Potential

**Tolga Kara** *  **and Ahmet Duran Şahin**

Sustainable Energy and Climate Systems Laboratory, Meteorological Engineering, Aeronautics and Astronautics Faculty, Istanbul Technical University, 34469 Istanbul, Turkey; sahind@itu.edu.tr
* Correspondence: karat19@itu.edu.tr

**Abstract:** This study examines the crucial role of wind energy in mitigating global warming and promoting sustainable energy development, with a focus on the impact of climate change on wind power potential. While technological progress has facilitated the expansion of the industry, it is crucial to continue making advancements to reduce the life-cycle emissions of wind turbines and ensure their long-term sustainability. Temporal discontinuities present a significant challenge for renewable energy sources. This study highlights the potential of hybrid systems to provide consistent energy output from wind sources. It also examines the variability in wind patterns caused by climate change, acknowledging that outcomes vary depending on geographic contexts, modeling approaches, and climate projections. Notably, inconsistencies in wind speed projections from downscaled general circulation models introduce uncertainties. While specific regions, such as North America, project an increase in wind speeds, others, such as the Mediterranean, face a potential decrease. Of particular note is the forecast for a potential long-term increase in wind speeds in Northern Europe. In conclusion, the wind energy industry displays considerable potential for growth, driven by technological advancements. However, the complexities resulting from climate change necessitate further research. Such insights are crucial for informed energy policy formulation and sustainable industry progress.

**Keywords:** climate change; hybrid system; wind energy; wind energy potential; wind speed

## 1. Introduction

Climate change refers to long-term alterations in typical weather patterns that define both local and global climates. Human activities, such as the burning of fossil fuels, deforestation, and agriculture, are the most commonly recognized factors contributing to climate change. These activities increase the amount of greenhouse gases, such as carbon dioxide, in the atmosphere. The enhanced greenhouse effect has led to global warming and an increase in the average surface temperature, especially since the Industrial Revolution in 1750, also known as the preindustrial era [1].

The aim of the 2015 Paris Agreement is to limit global warming to 1.5 degrees Celsius above preindustrial levels and to prevent it from exceeding 2 degrees Celsius [2]. This threshold is significant because scientific studies suggest that exceeding it could have serious and irreversible consequences for both ecosystems and human societies. These effects comprise of heightened frequency and severity of extreme weather events, elevated sea levels, decreased biodiversity, compromised food security, and perturbations to economic and social frameworks [2]. There are multiple ways to reduce greenhouse gas (GHG) emissions, including improving energy efficiency in all sectors, implementing low-carbon and renewable energy technologies, implementing carbon capture and storage techniques, and making changes to land use, among other strategies [3]. To achieve emission reductions, it is crucial to implement policy and regulatory measures such as carbon pricing, emissions trading systems, and regulatory standards [4]. In recent years, research in the fields of climate change and wind energy has significantly increased. Review articles play a crucial

role in investigating this extensive and ever-changing area, as they provide comprehensive analyses of wind energy trends and progress [5,6], the effects of climate change on wind energy [7], renewable energy [8], renewable energy supply [9] and energy sector vulnerability to climate change [10]. These analyses provide a comprehensive summary of current research and shed light on variations in approach and findings across various studies. The authors also offer informative perspectives on emerging trends and challenges, including the integration of wind energy into smart grids [11] and quantifying effects of climate change and extreme climate events on energy systems [12].

Climate change and wind energy have a bidirectional relationship. Firstly, the transition to wind energy is a crucial strategy for mitigating greenhouse gas emissions that drive climate change. Second, climate change can affect the potential and efficiency of wind energy by altering wind patterns. In this paper, we aim to expand on previous reviews by investigating the correlation between climate change and wind energy in greater depth. This review aims to answer two central questions:

(1)    What strategies, technological advances, and innovations have been used to enhance the wind energy industry and tackle the challenges presented by climate change?
(2)    How does climate change affect the potential of wind energy?

Alterations in atmospheric dynamics caused by climate change have a potential impact on the availability and predictability of wind resources, which subsequently affects the prospects of wind energy generation. Understanding climate-induced geographical and temporal variations in wind patterns is crucial for wind energy planning, development, and operation. It is important to consider these changes when evaluating wind energy projects. The impact of climate change on wind patterns affects the potential for wind energy generation. Recent studies have correlated rising global temperatures with changes in wind speeds in various regions. This interaction presents challenges and opportunities for the wind power industry, underscoring the need for adaptable techniques for effectively harness wind power amidst shifting climate conditions. It is essential to understand the synergies and challenges associated with climate change and wind energy to guide policy, planning, and investment decisions, particularly since they intertwine with broader energy systems.

### 1.1. Impact of Climate Change on Energy Systems

Climate change has wide-ranging effects on energy systems, impacting both production and consumption. As global temperatures rise, there is a higher demand for cooling services, which, in turn, leads to an increased need for electricity. Isaac and van Vuuren's projections propose that the need for residential air conditioning will rise by 72% by the end of the century in a high-warming scenario [13]. Although the decrease in the need for heating during milder winters appears logical, it is anticipated that the total energy demand will rise in temperate regions, owing to the increased necessity for air conditioning [14].

Climate change poses supply challenges, as higher temperatures decrease the effectiveness of thermal power plants that require more water to cool [15]. Renewable energy sources could also be affected. Changes in precipitation and reduced snowfall can affect hydroelectric power [16]. Wind power generation can also be affected by changes in wind patterns [17]. In addition, changes in cloud cover can affect solar energy production [18,19]. The potential consequences of extreme weather events combined with rising sea levels are of particular concern for energy infrastructure, especially in regions that are susceptible to storms, floods, or wildfires [20]. Therefore, strategic planning and policy formulation are crucial in the energy sector. To address the challenges posed by climate change, two main strategies are necessary: mitigation and adaptation. Mitigation aims to limit future global warming by reducing greenhouse gas emissions through improved energy efficiency, reduced use of high-emission services, and the promotion of low-carbon energy sources [3]. Adaptation involves responding to the effects of climate change on social, economic, and natural systems that are already occurring or are expected to occur in the future. This can involve actions at the local to national level, such as building flood defenses and modifying agricultural practices to adapt to new environmental realities [3]. In the energy industry,

taking a comprehensive approach to life cycle assessment (LCA) is critical. This technique evaluates the complete environmental effects of energy production, beginning with the extraction of raw materials and ending with disposal. The energy sector, which requires substantial amounts of water, faces significant obstacles due to droughts that can be amplified by climate change. Replacement of water-intensive thermal energy sources, with renewable sources such as wind and solar energy, can reduce the volume of water used for energy production [21].

When assessing the effects of different energy sources, wind energy emerges as a sustainable solution with low impact. Wind power's minimal water requirements, low emissions, and ability to bolster system resilience and security make it a key component in climate change adaptation strategies. Moving to renewable energy solutions, such as wind energy, can effectively reduce harmful air pollutants such as nitrogen oxides, sulfur dioxide, and particulate matter. These pollutants not only contribute to climate change but also have a detrimental effect on health [22]. Reducing air pollution yields substantial cost savings by mitigating healthcare expenditures, thereby reinforcing the economic argument for renewable energy [23]. Consequently, strategic planning and policy formulation are imperative in the energy sector to address and adapt to these challenges. As we move towards a more complete understanding of the broader effects, it is essential to examine the precise role of the energy sector in the mitigation of climate change, especially the emissions of greenhouse gases, which are fundamental to climate change.

### 1.2. The Role of Energy in Climate Change

The energy sector plays a significant role in climate change due to the emission of greenhouse gases during energy production and consumption. The combustion of fossil fuels, including coal, oil, and natural gas to generate electricity and heat, accounts for the largest share of global greenhouse gas emissions. In 2023, Liu reported that the energy sector was responsible for approximately 39.3% of the total global greenhouse gas emissions in 2022, based on real-time data from the Carbon Monitor project [24]. Human activities primarily release carbon dioxide ($CO_2$), the most dominant greenhouse gas, resulting from burning fossil fuels. The Global Carbon Project reports that in 2022, global $CO_2$ emissions from fossil fuel combustion and land use amounted to approximately 36.6 gigatons ($GtCO_2$), representing a 0.8% increase from the previous year due to fixed land use emissions and increased fossil $CO_2$ emissions. However, total $CO_2$ emissions remain lower than their 2019 levels and have essentially remained the same since 2015 [25–27]. In global debates on climate change, the focus is often on $CO_2$, which is identified as the primary greenhouse gas resulting from anthropogenic activities. $CO_2$ not only drives climate change, but several other greenhouse gases contribute to global warming as well. This includes methane, nitrous oxide, and trace gases such as the "F gas" group, which have also had a significant impact [28].

Fossil fuels still dominate global electricity production, accounting for 61% of the total. The impressive expansion of low-carbon wind and solar energy has resulted in a reduction in the usage of fossil energy. Currently, the overall contribution of wind and solar energy to worldwide electricity generation has exceeded 12%. This is more than twice its share since the agreement of the Paris Climate Accord in 2015 [29]. Despite this, coal still remains the largest source of electricity, responsible for 36% of global power generation and a quarter of all $CO_2$ emissions. Clean electricity sources now generate 39% of global electricity, with hydroelectric power accounting for 15%, nuclear power for 9%, wind power for 7.6%, and solar power for 4.5% [30]. In 2022, the intensity of carbon emitted from electricity generation reached a record low of 436 $gCO_2/kWh$ [29]. Figure 1 shows the carbon intensity in $gCO_2/kWh$ for the 20 countries with the highest electricity production in 2022. Furthermore, the Climate Action Tracker reported in its 2020 targets a carbon intensity of electricity generation of 50–125 $gCO_2/kWh$ in 2030, 5–25 $gCO_2/kWh$ in 2040, and almost zero in 2050 [31].

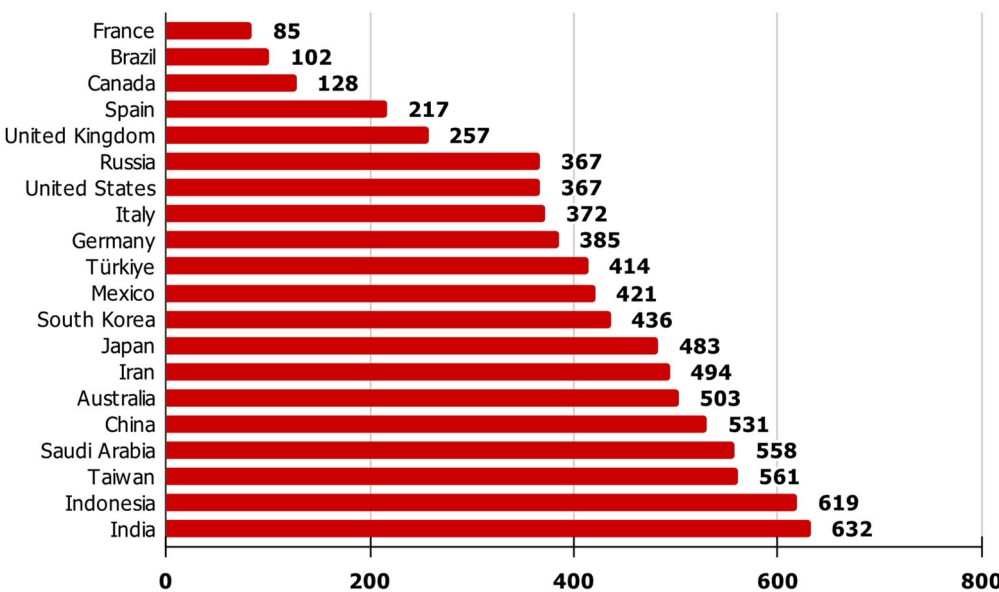

**Figure 1.** Carbon intensity in $gCO_2/kWh$ for the 20 countries with the highest electricity production in 2022 (Figure is regenerated from Ember data) [29].

In contrast, wind energy is a form of renewable energy that has minimal greenhouse gas emissions related to its production (see Table 1). Once installed, wind turbines do not emit greenhouse gases during operation. A comprehensive life cycle assessment (LCA) of wind turbines, which evaluates emissions from manufacturing, installation, maintenance, and eventual decommissioning, indicates that greenhouse gas emissions per unit of produced electricity are significantly lower compared to fossil fuel-based energy sources [3,32]. Wind power sets itself apart from other energy production methods in terms of emissions throughout its life cycle. The production and installation of all energy sources produce emissions, but, during operation, wind turbines do not emit pollutants. This stands in sharp contrast to electricity generation from fossil fuels, which releases significant amounts of greenhouse gases and other pollutants over its entire life cycle [32].

**Table 1.** Greenhouse gas emissions ($gCO_2/kWh$) from electricity supply technologies [3].

| Technology | Min | Median | Max |
|---|---|---|---|
| Coal—PC | 740 | 820 | 910 |
| Gas—combined cycle | 410 | 490 | 650 |
| Biomass | 130 | 230 | 420 |
| Solar PV—Utility-scale | 18 | 48 | 180 |
| Solar PV—rooftop | 26 | 41 | 60 |
| Concentrated solar power | 8.8 | 27 | 63 |
| Geothermal | 6 | 38 | 79 |
| Hydropower | 1 | 24 | 2200 |
| Nuclear | 3.7 | 12 | 110 |
| Wind Offshore | 8 | 12 | 35 |
| Wind Onshore | 7 | 11 | 56 |

Wind energy has a notable advantage from a life cycle assessment perspective due to its low water consumption. Unlike thermoelectric power generation, which relies heavily on water for cooling, wind energy does not require water usage [21]. This attribute makes it an attractive solution in a world where water scarcity is expected to become increasingly critical. However, it is crucial to acknowledge the lifecycle greenhouse gas (GHG) emissions of wind turbines and to continuously reduce them to ensure the sustainability of the wind energy sector. As the wind energy industry expands and technology advances,

the emissions associated with the life cycle of wind turbines are expected to decrease. This is attributable to advances in turbine efficiency leading to enhanced energy output and progress in the manufacturing process, resulting in decreased energy usage and emissions [33].

Several sectors of the economy contribute to greenhouse gas emissions, each with varying degrees of impact. By 2022, the power sector is expected to account for around 40% of global greenhouse gas emissions. Emissions arise primarily from the burning of fossil fuels, such as coal and natural gas, for the production of electricity and heat [24,25]. The transportation industry is projected to be responsible for approximately 24% of global greenhouse gas emissions by 2022. Road vehicles, aviation, and marine transportation are responsible for emitting a significant amount of greenhouse gases [24]. The combustion of fossil fuels, specifically gasoline in vehicles, is the primary source of these emissions. Ground transportation accounts for a proportion of 17.9%, with international air travel and shipping at 3.1% and domestic air transportation at 0.9% [26]. Moreover, industrial practices, including cement production and steel manufacturing, generate a substantial amount of greenhouse gas emissions. In 2022, the sector accounted for nearly 29% of total global emissions [24]. The production of cement releases $CO_2$ during limestone calcination, whereas steel production emits $CO_2$ during iron ore reduction [27]. The building sector is a significant contributor to global GHG emissions, primarily due to energy use for heating and cooling. In 2022, residential buildings were responsible for approximately 10% of overall greenhouse gas emissions, primarily due to electricity consumption and direct combustion of fossil fuels [24,28].

Understanding the relationship between the power and industrial sectors is pivotal. Both are deeply interconnected, with industrial processes relying on electricity and significantly affecting building emissions [33]. Such interdependency highlights the need for an inclusive strategy to mitigate greenhouse gas emissions. Progress in one sector can propel advancements in the other. Given the significant impact of the energy sector on climate change, it is vital to explore how transitioning to renewable energy sources, with a focus on wind energy, can improve energy security and resilience after the climate crisis.

## 2. Materials and Methods

To fully comprehend the effects of climate change on wind energy's capabilities, a comprehensive investigation of the literature was conducted from January 2021 to June 2023. Relevant literature was collected from well-regarded academic databases such as Science Direct, Scopus, and Google Scholar, along with reports from respected organizations such as the European Union, WindEurope Association, Ember, International Energy Agency, and International Renewable Energy Agency. The search terms and keywords used included "wind energy", "wind potential", "climate change impacts", "renewable energy", "hybrid systems", and "climate modeling". The review inclusion criteria included articles, books, conference proceedings, and authoritative reports that explicitly addressed the impact of climate change on wind energy. In contrast, the excluded criteria were articles that did not have peer reviewed, outdated studies (before 2000) and publications that were not related to the central theme of the review. Sources that met the inclusion criteria were distilled for their key findings, methods, and conclusions.

To assess the influence of climate change on the potential of wind energy, studies using general circulation models were analyzed. These models, both statistically and dynamically downscaled, provided predictions of future wind speeds. To gain a comprehensive understanding of the impact of climate change on wind energy, the results of various models and scenarios were compared.

## 3. Results

### 3.1. Wind Energy and Its Role in the Global Energy Mix

Wind energy plays an increasingly crucial role in the global energy portfolio. With its recognition as an environmentally friendly, renewable, and economically viable source of

electricity, it is becoming a promising option. According to the IEA, it has rapidly emerged as the leading source of renewable energy worldwide and accounts for approximately 17% of global electricity generation in 2021, a figure that is expected to increase further [34]. Factors driving the adoption of wind energy include decreasing costs [35], advances in wind technology, and a higher demand for low-carbon power sources.

Wind energy directly reduces greenhouse gas emissions and plays a pivotal role in mitigating climate change. Wind energy also benefits the economy by creating jobs and stimulating local economic growth. The expansion of the wind energy sector provides considerable employment prospects in the manufacture, installation, maintenance, and operation of wind turbines and farms [36]. However, depending on the availability, infrastructure capabilities, and policy support of wind resources, the role and impact vary considerably between regions. A complete understanding of the contribution of wind energy to the global energy mix requires a comprehensive investigation of regional inequalities [37]. Integrating wind power into the power grid is challenging due to its variability. A more comprehensive understanding of the role of wind energy in the energy mix can be obtained by researching probable solutions, such as energy storage and grid flexibility [38–42].

Future projections indicate the increasing importance of wind energy. According to recent estimates from IRENA and the IEA, wind energy could potentially supply between 25% and 33% of the world's electricity by 2050, highlighting the crucial role it can play in the global energy mix [43,44]. Government policies play a vital role in promoting the use of wind energy. Some examples of such initiatives are feed-in tariffs, power purchase agreements (PPA), tax incentives, and renewable portfolio standards. Further examination of policy implementations around the world can provide valuable information on the most effective strategies to promote the deployment of wind energy [45].

Furthermore, international agreements, such as the Paris Agreement, reaffirm the commitment to embrace renewable energy, including wind energy. The shared goal of containing global warming has motivated global action and underscored the crucial role of renewable energy in achieving this objective. Wind energy projects demonstrate significant scalability, ranging from community-based wind farms to large offshore installations [46–50]. Given effective regulatory frameworks and supportive policies, each country has the ability to customize the implementation of wind energy projects to address unique energy requirements and contexts. Despite recent progress in adopting wind energy, there remain hurdles. Regulatory complexity, infrastructure and grid integration, social acceptance, and the intermittent nature of wind energy require additional investigation and solution-focused research [51–56].

Wind energy plays a multidimensional role in the global energy mix, having an impact on environmental, economic, and social aspects. As a renewable and clean energy source, wind energy is a vital element in worldwide efforts to move towards low-carbon economies. Given the urgent need to reduce the impact of climate change, it is probable that future reliance on wind energy will increase, supported by ongoing progress in technology, policies, and environmental regulatory standards.

### 3.2. Enhancing Resilience and Energy Security

Reducing carbon emissions from electricity generation is crucial to limit global warming to 1.5 °C. To achieve this, we need to quickly transition to renewable energy sources and the rapid phasing out of coal, oil, and natural gas [57]. This transition is essential to effectively reduce greenhouse gas emissions, in accordance with the objectives of the Paris Agreement [58]. The shift from fossil fuels to renewable sources of energy is a pivotal aspect of the transformation of energy and covers significant changes in energy systems. This transition is stimulated by advances in technology, regulatory frameworks, economic incentives, and evolving market conditions [59]. Wind energy has been a prime example of these developments, manifesting its potential to enhance the resilience and security of energy systems through technological progress and practical applications. Diversification

of the energy portfolio, decentralization of production, and innovation are key to realizing the benefits of wind energy for a more sustainable energy future.

Integrating wind energy into the energy mix offers the potential to significantly reduce dependence on foreign oil and improve energy security. In particular, countries like Denmark and Germany serve as examples in this regard by seamlessly integrating wind energy and reducing their dependence on fossil fuels [60]. Distributed wind energy systems improve grid resilience through their design. By decentralizing power generation, such systems are less vulnerable to local disruptions. As extreme climate changes become more frequent, distributed systems are becoming more resilient by minimizing transmission losses and improving dependability [61].

Technological advancements in the wind industry, specifically in energy storage and advanced grid management, are vital to integrate a higher percentage of variable renewable energy sources into the grid [38,39,62,63]. The International Energy Agency stresses the importance of these innovations, particularly in creating state-of-the-art energy storage systems [59,60].

It is essential to acknowledge that while mitigating and adapting to climate change risks are necessary, they serve different purposes within the energy sector. Mitigation involves the shift towards low-carbon energy sources, whereas adaptation requires modifications in infrastructure design and operation to deal with climate variables. Both aspects are essential to address the multifaceted consequences of climate change [64,65]. Although the transition to renewable energy sources, such as wind power, is crucial to improving resilience and securing energy, the driving forces or impediments to this transition are heavily influenced by technical policies and regulatory frameworks. The following section discusses numerous technical policies and their effects on the wind energy industry.

### 3.3. Technical Policies for Environmental Protection and Their Implications for Wind Energy

Technical government policies have been critical in stimulating the transition to a low-carbon energy system. These policies include performance standards, emissions trading systems, and feed-in tariffs. They aim to mitigate the impact of climate change by reducing greenhouse gas emissions, promoting innovation in renewable energy technologies, particularly wind energy, and ensuring consistent growth in the wind energy sector despite climate change [34,35]. Therefore, performance standards are a crucial factor in driving technological innovations in the wind energy sector. They guarantee that wind turbines and related technologies meet stringent efficiency requirements, induce innovation, and ensure that the wind energy industry continues to be a leader in sustainable energy solutions.

Emissions trading systems, or cap-and-trade systems, indirectly promote wind energy by raising the cost of carbon-intensive energy sources, making wind energy more competitive. However, challenges arise from complex permit allocation, market instability, and government regulation, which can affect the competitiveness of wind energy across industries and regions [37,50]. If permits are allocated in a way that favors carbon-intensive industries, it can hinder the growth of renewable energy sources such as wind. This can lead to lower carbon prices, making wind energy less competitive in the energy market. Also, emissions trading schemes can introduce volatility into the carbon market, affecting the cost of emission allowances. This market instability can create uncertainty for wind energy investors, making it difficult to plan and finance projects. Sudden changes in government policy or economic conditions can cause carbon prices to fluctuate. For wind energy projects that rely on revenue from the sale of emission allowances, these fluctuations can affect project profitability and financial viability.

Feed-in tariffs, which guarantee a fixed price for renewable energy fed into the grid, have been particularly effective in accelerating the deployment of wind energy in various regions [41]. Although these tariffs have significantly increased wind energy in some regions, they have also received criticism. Market distortions unique to the wind energy industry have been a topic of discussion. Concern has been raised about excessive investment in areas that already have sufficient wind infrastructure, as well as the long-term financial

stability of these tariffs [49]. This oversupply of renewable energy can cause imbalances in the energy market, which can affect pricing and competition. In certain regions, the surge in wind energy projects supported by feed-in tariffs has resulted in an excess of renewable energy at certain times of the day. This oversupply can decrease energy prices, jeopardizing the profitability of renewable energy projects and potentially causing financial strain for investors. Another concern centers on the long-term financial stability of the feed-in tariff programs. Such programs often require long-term contracts with fixed prices, which can create financial burdens for governments or utilities as energy production increases over the years. Balance the need to stimulate renewable energy while ensuring financial responsibility is a delicate task. A region that initially offered generous feed-in tariffs to attract wind energy investments may find itself facing increasing financial commitments as the number of wind projects grows. The financial feasibility of these initiatives while meeting the objectives of renewable energy requires careful planning and routine modifications.

The importance of wind energy in achieving global environmental goals is substantial. For example, in 2018, wind energy made a significant contribution to the reduction in global $CO_2$ emissions, underscoring its potential to offset greenhouse gas emissions [58]. As nations work to fulfill their obligations under the Paris Agreement and other international agreements, the importance of wind energy is increasing [43]. Countries have utilized the nationally determined contributions (NDC) outlined in the Paris Agreement as a reference to strengthen their renewable energy portfolios. For example, the European Union acknowledges that wind energy can achieve its goal of reducing greenhouse gas emissions by 55% by 2030 [44]. Similarly, the climate action strategies of the United States, China, and India have incorporated wind energy, underlining its significance in creating a future of sustainable energy [45–47]. In any case, the shift to a future that depends on wind energy poses its own set of difficulties. When drafting technical guidelines, socioeconomic factors, technological constraints, and political circumstances should be considered [48]. Although feed-in tariffs have effectively promoted wind energy in various places, they have received criticism for creating financial instability and market distortions [49]. Likewise, despite their theoretical effectiveness, emission trading systems have faced problems concerning permit allocation, regulatory oversight, and market volatility [50]. The interplay of technical policies, wind energy, and climate change underscores the need for a comprehensive and thoughtful approach to policy development to ensure sustainable growth in the wind energy sector amid changing climate patterns. Technical policies serve as a means to incorporate wind energy into our systems.

### 3.4. Climate Scenarios and Implications for Wind Energy

Projections of future climate reveal complex interconnections between human activities, emissions of greenhouse gases, atmospheric composition, and resulting alterations in temperature patterns. These projections, which encompass multiple scenarios and goals, serve as a fundamental framework for understanding the future course of climate dynamics and their considerable effects on renewable energy sources, particularly wind power [3,66].

The standardization of climate scenarios has been a significant undertaking. In 1990, the IPCC released the Special Report on Emission Scenarios (SRES), also known as the SA90 scenarios [67]. Subsequently, the implementation of the IS92 emission scenarios in the mid-1990s provided a broader range of emissions options, resulting in a more comprehensive collection of potential future climate scenarios [68]. The standardization of climate scenarios has been a major effort. It began in 1990 with the launch of the IPCC Special Report on Emissions Scenarios (SRES), known as the SA90 scenarios [68]. In 2010, the Representative Concentration Pathways (RCPs) were introduced. [69]. While these scenarios allow for a variety of emission scenarios, making a prompt transition to renewable energy sources is unquestionable. Wind energy is emerging as a critical factor in reducing the detrimental impacts of climate change [70]. It is aligned with lower emission pathways, including B1 and B2 scenarios [66]. To achieve these goals, further investment in this renewable energy source is necessary.

The Representative Concentration Pathways (RCPs) provide crucial insights into climate stabilization [3]. These pathways, denoted by numerical values such as RCP 2.6, RCP 4.5, RCP 6.0, and RCP 8.5, represent distinct emissions trajectories and radiative forcing levels. Notably, RCP 2.6 places strong emphasis on renewable energy, particularly wind energy, within this low-emission trajectory [71]. The sixth IPCC Assessment Report (AR6) achieved a milestone by introducing Shared Socioeconomic Pathways (SSP), which integrate RCPs with socioeconomic factors to provide clearer insights into future climate trajectories [72,73]. Within this context, the SSP framework highlights the importance of renewable energy sources, specifically with regard to wind energy, to bring about the desired climate outcomes. These scenarios suggest that it is necessary to increase both offshore and onshore wind energy capacities to combat rising temperatures and meet sustainability goals [74,75]. International climate policies are key in unlocking the complete potential of wind energy to mitigate the consequences of climate change, given the different implications that arise from various climate scenarios.

*3.5. Developing an International Climate Policy to Protect the Environment*

The persistent increase in human-induced greenhouse gas emissions, primarily due to the Industrial Revolution, has increased global temperatures by approximately 1 °C [72]. Tackling this issue requires a collaborative partnership on a global scale, with emphasis placed on renewable energy options, specifically wind energy, which assumes a significant role in influencing international policy paradigms [76]. Facing global warming and its far-reaching consequences, international cooperation is mandatory. In this regard, the usage of wind energy has become indispensable in sustainable global policies.

The United Nations Environment Program, initiated in 1972, marks the beginnings of the global efforts to address environmental issues. The First World Climate Conference, which took place in 1979, served as the starting point of discussions centered around climate change. Milestones such as the Villach Conference in 1985 and the Brundtland Commission's Our Common Future report emphasized the urgency of addressing greenhouse gas emissions with renewable energy sources playing a critical role [77]. International agreements, including the Vienna Convention in 1985 and the Montreal Protocol in 1987, further reinforced the consensus in favor of embracing renewable energy sources. The Toronto Conference in 1988 was particularly notable for setting emission reduction targets and for promoting the use of wind energy. The UN Conference on Environment and Development (UNCED) in 1992 established the Framework Convention on Climate Change (FCCC), setting the stage for the Kyoto Protocol in 1997. The protocol emphasized the use of renewable energy, particularly wind power, to address climate change [6]. The 2015 Paris Accord clearly articulated the need to limit global temperature rise to below 2 °C, with wind power positioned as a transformative solution. This landmark agreement established nationally determined contributions (NDCs), with countries integrating wind power into their climate change mitigation plans [78].

Future research and policy discussions must focus on enhancing technical regulations while considering stakeholder views. A considered approach that balances economic, environmental, and societal considerations is essential as we navigate toward a low-carbon, sustainable future. This study examines the link between climate change trends and wind energy's viability, determining the diverse climate change factors that impact the feasibility of wind energy across regions.

*3.6. Current Status of Wind Energy*

Wind energy is an increasingly important player in the global energy sector. According to [35], the total installed capacity of global wind power will reach 960 GW by 2022. Europe, particularly Germany and Spain, as well as North America, with the United States leading the way, are the primary contributors to this development. China leads the world in installed wind power capacity, with more than 365.5 GW by 2022, followed by the United States (144.2 GW) and Germany (67 GW) [35].

The global wind energy sector is experiencing both growth and challenges. In China, onshore wind installations bounced back in 2022, reaching 32.5 GW. This rebound was driven by the approval of over 60 GW of onshore wind capacity under the "grid parity" scheme in 2021 [35,79,80]. This shows the commitment to achieving ambitious renewable energy goals, even with possible grid connection challenges. In contrast, the US onshore wind industry experienced slower growth in 2022. While the Internal Revenue Service (IRS) extended the production tax credit (PTC) for projects started in 2016 or 2017, the COVID-19 pandemic, as well as the supply chain and grid interconnection problems, led to a 32.4% decrease in quarterly installations compared to 2021 [35]. India experienced obstacles in the commissioning of wind power, such as high inflation, unavailability of the grid, and delays in the timeline. Although they commissioned 1.847 GW of wind power by 2022, the overall increase was less than expected [35]. On the other hand, Germany aimed to decrease its reliance on imported fossil fuels by raising its renewable objectives for 2030 and enacting the "Onshore Wind Law" in July 2022. Despite a slightly lower-than-expected number of onshore wind installations in 2022, Germany maintained its position as Europe's largest wind market for new additions in the same year [35]. Although the COVID-19 pandemic and political instability, Brazil's wind industry demonstrated resilience by reaching a record high of more than 4 GW of offshore wind power plants in 2022, thanks to public auctions and private power purchase agreements (PPA). However, in Vietnam, more than 1 GW of onshore wind projects encountered delays because there was no ceiling price to negotiate PPAs with investors for renewable projects until January 2023 [35].

Significant progress in offshore wind energy is demonstrated by global capacity, which surpassed 35 GW in 2021, generated by offshore wind energy in 2021, with Europe leading the way with a capacity of 22 GW, followed by Asia with 10 GW. The remaining 3 GW were distributed in other regions [35]. These figures indicate the developing global trend towards sustainable offshore wind energy as an alternative power source. The evolution of offshore wind energy technology is essential to its growth, with turbines becoming larger, more efficient, and even capable of floating now [81,82]. These advancements have expanded the accessibility of offshore wind energy, making it a viable prospect for numerous regions around the globe. Nevertheless, with progress come obstacles. The offshore wind industry has substantial expenses for installation and maintenance. Furthermore, the power generated into the grid poses significant challenges. The sector must also face the environmental and social ramifications associated with offshore wind farms [81–85]. Dedicated research and policy interventions are necessary to ensure sustainable growth in the offshore wind power sector.

Offshore wind energy has made progress at varying levels. In 2022, Hornsea Project 2 in the UK fully commissioned all of its offshore turbines, whereas the Seagreen Project in Scotland faced delays and only connected 27 wind turbines (255 MW) to the grid. Germany has had relatively low offshore wind installations since 2020, primarily due to unfavorable offshore wind policies and a limited short-term project pipeline [35]. Meanwhile, China's offshore wind installations saw a significant decrease after the implementation of "grid parity" in the country's offshore wind market in 2022. This is despite a record-breaking year in 2021, during which almost 17 GW of offshore wind power were connected to the grid [35]. The future of this sector is promising given the continuous technological innovations and the declining costs. These factors naturally position the sector to play a more important role in the future energy mix. Offshore wind energy has enormous potential in the future, with the International Energy Agency (IEA) projecting a $1 trillion industry by 2040 and a worldwide capacity of up to 420 GW [86]. Offshore wind energy plays a crucial role in the renewable energy sector. However, it is also important to consider onshore and other renewable sources of energy to better understand their importance in the overall energy mix. The industry must address numerous challenges, and its ability to navigate these obstacles will significantly impact its role in shaping the future of renewable energy.

Wind energy projects use various types of turbines, including offshore and onshore turbines. While onshore turbines are more widely used, they have capacities ranging from 1

to 4 MW, with rotor diameters of 50 to 140 m. Offshore wind turbines, although pricier, can capture more powerful winds and generate more electricity [6]. Technological advancements have led to larger and more efficient turbines, with contemporary projects featuring capacities ranging from 3 to 4 MW onshore and 8 to 12 MW offshore. The amplified capacity of turbines has substantially increased the potential for wind energy production.

Over time, the cost effectiveness of wind energy has improved significantly. Since 1980, wind energy has decreased by approximately 70%, allowing it to compete with conventional energy sources in numerous regions worldwide. Onshore wind power is one of the most cost-effective electricity sources, with costs that vary depending on the project and region, ranging from $20 to $50 per MWh [87]. Even though the prices for commodities and renewable equipment have increased, in 2021, newly commissioned solar photovoltaic (PV), offshore, and onshore wind power projects experienced a decrease in electricity costs on a global scale. These cost reductions were driven by better performance and enhanced capacity factors, specifically in the offshore wind (Table 2).

**Table 2.** Global weighted average total installed cost, capacity factor, and levelized cost of electricity trends by renewable technologies, 2010–2021 [87].

| | Total Installed Costs (2021 USD/kW) | | | Capacity Factor (%) | | | Levelized Cost of Electricity (2021 USD/kWh) | | |
|---|---|---|---|---|---|---|---|---|---|
| | 2010 | 2021 | Percent Change | 2010 | 2021 | Percent Change | 2010 | 2021 | Percent Change |
| Bioenergy | 2714 | 2353 | −13% | 72 | 68 | −6% | 0.078 | 0.067 | −14% |
| Geothermal | 2714 | 3991 | 47% | 87 | 77 | −11% | 0.050 | 0.068 | 34% |
| Hydropower | 1315 | 2135 | 62% | 44 | 45 | 2% | 0.039 | 0.048 | 24% |
| Solar PV | 4808 | 857 | −82% | 14 | 17 | 25% | 0.417 | 0.048 | −88% |
| CSP | 9422 | 9091 | −4% | 30 | 80 | 167% | 0.358 | 0.114 | −68% |
| Onshore wind | 2042 | 1325 | −35% | 27 | 39 | 44% | 0.102 | 0.033 | −68% |
| Offshore wind | 4876 | 2858 | −41% | 38 | 39 | 3% | 0.188 | 0.075 | −60% |

The global weighted average levelized cost of electricity (LCOE) for new onshore wind projects decreased by 15% compared to the previous year, reaching USD 0.033/kWh in 2021. Onshore wind projects outside of China also saw a decrease in costs, 12% falling to USD 0.037/kWh. China took the lead in new onshore wind capacity and experienced price reductions for wind turbines. The offshore wind market saw substantial growth in 2021, adding 21 GW of new capacity and experiencing a 13% year-over-year decrease in the global weighted average cost of electricity for offshore wind, reaching USD 0.075/kWh [87].

From 2010–2021, there has been a significant change in competitiveness between renewable energy and traditional options, such as fossil fuels and nuclear power. The weighted average LCOE of solar PV projects globally decreased by 88% during the period, whereas onshore wind and Concentrated Solar Power (CSP) projects experienced a reduction of 68%, and offshore wind projects witnessed a decrease of 60% in costs [87]. However, despite significant growth and encouraging trends, there are several challenges. The intermittent nature of wind can cause stability issues while integrating substantial amounts of wind power into the grid. While energy storage technologies could offer solutions to world energy challenges, they also face significant hurdles, such as high costs and technical issues [40]. Social acceptance and environmental factors, such as noise and the impact on wildlife, also pose considerable obstacles [88]. Nevertheless, wind energy shows promising growth and significant technological and economic advancements. However, to fully exploit the potential of wind energy and meet global energy demand sustainably, several challenges need to be addressed [54].

*3.7. Intermittency and Hybrid Energy Systems*

The intermittent nature of wind energy, a major challenge in renewable energy systems, creates instability in production and reliability problems [89]. This can lead to grid

instability and challenges in balancing electricity supply and demand [51,63]. One of the potential solutions to these discontinuities is the application of hybrid energy systems. This section will go into the analysis of various hybrid energy systems, detailing their technical and economic feasibility and their potential to address these challenges.

Hybrid energy systems, which combine two or more energy sources, have attracted interest because of their potential to provide a more stable and reliable power supply. These systems seek to complement wind energy with other renewable sources with different or opposing variability patterns, thereby mitigating the intermittency of wind–solar hybrid systems, for example, by taking advantage of the complementary nature of wind and solar energy. Wind speeds are typically higher at night and during certain seasons, whereas solar energy production peaks during the day and is more abundant in other seasons. This inverse correlation can result in a more consistent energy output throughout the day and year [90,91]. The integration of battery backup systems into these hybrid systems can potentially stabilize their output, an idea that has been explored in several studies [63,91–93].

Another innovative solution is the combination of offshore wind turbines with floating photovoltaic (FPV) systems, which has been shown to have a 68% reduction in variability of power output compared to a standalone wind farm [94]. Similarly, hybrid wind–hydro systems, where surplus electricity from wind energy production is used to pump water uphill into a storage reservoir, can ensure a more constant power supply [95,96]. Anagnostopoulos and Papantonis presented a successful example of such a system on the Greek island of Crete, where the hybrid system effectively recovered 40–60% of excess wind energy [95]. Taking into account advances in battery technology [97,98], wind–battery hybrid systems also offer a feasible solution to the discontinuity problem. Energy stored from periods of high wind speeds can be sent when wind power production is insufficient. Furthermore, in remote or isolated areas without grid connections, hybrid wind–diesel systems have proven particularly effective in regions such as Alaska, reducing operating and maintenance costs and mitigating $CO_2$ emissions [99]. Incorporating renewable energy resources into desalination plants [100,101], as demonstrated in the Kwinana desalination unit in western Australia, can potentially address the high energy consumption and environmental issues associated with desalination [102]. Although technical considerations are paramount, the economic feasibility of hybrid energy systems is also critical. Although hybrid systems often require higher initial investments compared to single-source systems, their potential to provide a more consistent power supply may improve their long-term economic viability, especially considering the social costs of power outages and grid instability [39,103].

Hybrid energy systems present promising solutions to the discontinuity of wind energy production. When wind power is combined with other energy sources, they offer a more stable and reliable power supply, although at a potentially higher initial cost. Future research and technological advances could further enhance the technical and economic feasibility of these systems, contributing to a more resilient and sustainable energy future.

### 3.8. Storage Technologies for Wind Energy

Efficient energy storage technologies are necessary to mitigate the variability and intermittency of wind energy production. These technologies store excess wind energy during high wind speeds for later use when the wind speeds decrease. They play a crucial role in the effective utilization of wind energy and improve the reliability and stability of wind power systems. Several energy storage options have been investigated, such as batteries, pumped hydro, and thermal storage. The selection of the most appropriate storage method depends on several factors, including storage capacity, discharge duration, wind farm location, and overall cost effectiveness of the technology.

Energy storage in wind energy systems has become increasingly popular, and lithium-ion batteries being a common option due to their high energy density, extended cycle life, and declining cost [104,105]. Tesla's Hornsdale Power Reserve project in South Australia has exemplified the feasibility of this approach by integrating a 100 MW/129 MWh lithium-

ion battery system with a wind farm [106]. However, the challenges posed by the need for regular battery replacement and limited battery life must be addressed.

Pumped hydro storage (PHS) represents another widely used technology for energy storage. In PHS, excess electricity pumps water from a lower reservoir to a higher one, which can then be released during periods of electricity demand to generate hydroelectric power. PHS systems feature large storage capacities and long discharge durations, making them suitable for utility-scale applications. However, the implementation of this solution is restricted by specific geographical conditions, namely, the availability of two reservoirs located at different heights, which ultimately limits its practicality [95,107–109].

Another potential solution for wind energy storage is thermal energy storage, which harnesses excess wind energy to heat a storage medium, such as molten salt or rocks, for later use. The stored thermal energy can then be converted back into electricity when necessary. Thermal storage systems have high energy storage capacities and efficiencies and are less reliant on specific geographic conditions compared to pumped hydro storage. However, the conversion process from thermal energy to electricity is less efficient, which can affect the overall efficiency of the system [110,111].

Flywheels, supercapacitors, and hydrogen storage are additional innovative solutions for energy storage that are currently under exploration. Flywheels and supercapacitors offer short-term energy storage with rapid charge and discharge rates, making them ideal for frequency regulation in wind power systems [112]. Hydrogen storage entails the utilization of surplus wind energy to create hydrogen through electrolysis, which can later generate electricity through the use of fuel cells or combustion, as required. Although energy storage technologies offer promising potential, their high costs and technical challenges currently limit their large-scale application [113,114].

The integration of energy storage systems with wind energy systems is a critical factor in improving the reliability and usability of wind power [115,116]. While each storage technology presents its own set of advantages and challenges, the selection of the most appropriate technology should account for the unique requirements and limitations of each wind energy project [117]. The ongoing advancements in energy storage technologies and cost reductions are poised to play a key role in facilitating the wider adoption of wind energy.

*3.9. Identification of Trends and Patterns in Wind Energy Potential under Various Climate Change Scenarios*

In examining the impact of climate change on wind energy potential, it is critical to navigate the complex array of findings. Contributing factors to this complicated situation include geographical location, utilized models, time periods, and considered climate scenarios. (see Table 3). Future wind speeds have been generally forecasted using statistical downscaling of general circulation models with mixed results. However, it is important to note that the impact of climate change on wind energy potential remains complex and location dependent, despite variations in findings.

To begin, this study examines the forecasting of future wind speeds through the utilization of statistical downscaling of general circulation models. It is noteworthy that these results display significant variability and limited agreement among the models. Eichelberger's research, which utilized CMIP3 models, indicates that North America may witness increased wind speeds in specific seasons. The utilization of these climate models provides valuable insights into potential future trends. According to Eichelberger's findings using CMIP3 models in Scenarios A2 (refers business as usual) and B1 (refers sustainable development and has half the $CO_2$ emissions of A2 in 2100) wind speeds are expected to increase in mean annual, winter, and summer in North America in 2050. This could potentially improve the production of wind energy in the region [118]. However, the situation is different when we explore the eastern Mediterranean, where the impact of climate change on wind energy takes a different turn. Bloom used the PRECIS regional climate model, developed by the Hadley Center, with the IPCC A2 emissions scenario simulation to demonstrate an increase in wind speeds over land but a decline over the

Mediterranean Sea in period of 2071–2100 [119]. This study reinforces the regional disparities, with wind speed changes over land differing from those over the Mediterranean Sea, except for the Aegean Sea. These regional nuances are further illuminated by studies conducted by Barstad [120], Carvalho [121], Davy [122], Emodi [123], Katopodis [124], and Koletsis [125], which provided broad agreement on the likelihood of reduced wind speeds in the Mediterranean region. Multiple studies concur on the projection of reduced wind speeds in the Mediterranean region, further emphasizing this consistent trend.

Carvalho noted that under the RCP 8.5 scenario, there could be an increase in wind speed in the Baltic Sea [121]. However, an exception to the trend is observed in the Baltic Sea, where wind speed may increase under the RCP 8.5 scenario. An interesting anomaly arises in the Baltic Sea, where wind speeds may deviate from the regional trend under the RCP 8.5 scenario. Several studies conducted by Hueging (2013) and Koletsis (2016) using ensemble projections from different regional climate models in the A1B scenario indicated that wind speed would increase in central and northern Europe while decreasing in the Mediterranean, excluding the Aegean Sea [125,126]. These studies collectively suggest a pattern of increasing wind speeds in central and northern Europe, contrasting with decreases in the Mediterranean, excluding the Aegean Sea. Hosking (2018) presented that UK wind patterns may experience substantial seasonal fluctuations that minimally impact annual wind power production. Interestingly, wind speeds could increase as a result of the northward migration of Atlantic jets under 1.5 °C forcing [127]. Near the shoreline of Croatia, Pasicko discovered a rising trend in wind speeds within that area [128]. Notably, a study indicates that wind speeds could increase due to specific climate phenomena. Additionally, Pasicko's research suggests a rising trend in wind speeds near the shoreline of Croatia, potentially impacting local wind energy production.

Through a rare comparison between measurement and simulation data, Katopodis determined that there were no substantial shifts in Greece's potential for wind energy by the year 2040, which is the expected lifespan of the installed wind turbines, based on RCP climate projections [124]. Furthermore, Jerez examined the impact of climate change on the combined production of wind and solar photovoltaic power over Europe. He found that wind energy outcomes in both northern and southern Europe were comparable [129]. These studies offer valuable regional insights. Katopodis' research suggests Greece's wind energy potential is stable, whereas Jerez's work reveals similar results in northern and southern Europe. In addition to Europe and the United States, studies by Fant and Schlosser in Southern Africa [130] and Chang in Taiwan [131] indicate varying impacts of climate change on wind energy beyond Europe and the United States. Thus, studies in Africa and Asia demonstrate diverse impacts of climate change on wind energy, highlighting the global scope of this issue.

The aforementioned studies suggest that certain areas may experience an increase in wind speeds, whereas others may face a decrease. Although projections from multimodel GCM ensembles indicate potential reductions in annual maximum wind speed over the United States [132], Europe [133] and northern Europe could witness an increase in wind speeds over the 50-year return period only by the end of the twentieth century [134]. Furthermore, changes in the extreme wind wave loading on offshore wind turbines, especially those associated with tropical cyclones [135], could be significant in regions with significant wind turbine deployments [136]. Moreover, Anagnostopoulou (2013) utilized the two-step cluster analysis method and determined that future patterns of Etesian wind could vary, but there would probably be no noticeable change in wind direction. In late 21st-century projections following the SRES A1B scenario, the intensification of Etesian winds is anticipated, and future projections indicate a decrease in subsidence phenomena over the eastern Mediterranean [137]. Additionally, the impact of climate change on extreme wind wave loading on offshore wind turbines is noted, particularly concerning tropical cyclones. Nevertheless, it is important to acknowledge that research in this domain is still limited and specific to regions. For example, one study projects an increase in wave height from the 20-year return period in the North Atlantic [138], and another suggests a decrease in

the western tropical Pacific [139]. As an example, one study anticipates an increase in wave height in the North Atlantic, whereas another predicts a decrease in the western tropical Pacific. In summary, the reviewed studies collectively indicate regional variations, with some areas experiencing increased wind speeds, whereas others may encounter reductions. Notably, northern Europe might see an increase in wind speeds in the long term.

The variability observed in these projections highlights the complex relationship between climate nonstationarity and wind energy production. While experts agree that climate change will affect wind speeds and thus wind energy production, the details of these changes—especially their location and magnitude—are still unclear. The various projections highlight the complex connection between non-stationarity in climate and the generation of wind power. While it is widely accepted that climate change will have an impact on wind speeds and energy production, the exact details of these changes—including their geographic distribution and magnitude—are still uncertain.

**Table 3.** Selected studies on the impacts of climate change on wind speed/potential/generation.

| Study Area | Scenarios | Historical Period | Future Period | Projected Changes | Ref. |
|---|---|---|---|---|---|
| **America** | | | | | |
| United States (US) | SRES A2 | 1968–2000 | 2038–2070 | Parts of Kansas, Oklahoma, and northern Texas are projected to possess greater wind energy potential. | [140] |
| United States | RCP 8.5 | 1979–1999 | 2079–2099 | Coastal regions within the United States may experience a 5–10% increase in 10 m wind speed per century based on climate models, but a 5–10% decrease in summer. | [141] |
| CONUS | SRESA1B | 1990 to 1999 | 2040 to 2049 and 2090 to 2099 | By the 2040s, the Great Plains, Northern Great Lakes Region, and Southwest US situated to the southwest of the Rocky Mountains can expect an increase in wind speeds ranging from 0.1 to 0.2 m/s. In the 2090s, the Great Plains Region and the Southwestern US will experience an overall increase in wind speed with a mean increase of 0 to 0.1 m/s. Nevertheless, some coastal areas in the US may experience decreasing summertime winds. | [142] |
| Brazil | RCP4.5 | 1961–1990 | 2021–2050 and 2070–2099 | Wind power potential at certain locations in Northeast Brazil could increase by over 40%. | [143] |
| United States of America (USA) | RCP 8.5 | 1980–2005 | 2075–2099 | Annual energy production increases by 8% in the Southern Plains and decreases by 5% in the Northern Plains. | [144] |
| America | SSP2-RCP2.6 and SSP2-RCP 6.0 | 1970–2000 | 2070–2100 | Regions that decrease in wind energy generation are Mexico and Central America, which are not statistically significant. | [9] |
| **Africa and Asia** | | | | | |
| Southern Africa | SRES A2 and B1 | 1979–2009 | 2050 | The median of the long-term mean of wind speed is expected to be close to zero by 2050. | [130] |
| South Africa | RCP4.5 and RCP8.5 | 1981–2005 | 2051–2075 | The average daily wind speeds in the northeast region of South Africa are forecasted to rise, but not exceeding 6%. This increase lies within a range suitable for power generation. Nonetheless, wind energy density is foreseen to persist at a low level. | [145] |
| West Africa | RCP4.5 and RCP8.5 | 1971–2000 | 2021–2050 and 2071–2100 | A reduction in energy production of up to 12% is expected in the near future, whereas power production is estimated to increase by approximately 24–30% over most regions in the far future. | [146] |
| Taiwan Strait | SRES A1B | 1981–2000 | 2011–2040, 2041–2070, and 2071–2100 | Wind resources in the eastern half of the Taiwan Strait have decreased by 3% in comparison to previous years. | [131] |

**Table 3.** *Cont.*

| Study Area | Scenarios | Historical Period | Future Period | Projected Changes | Ref. |
|---|---|---|---|---|---|
| America | | | | | |
| Japan | +4K warming | 1951–2010 | 2051–2110 | Wind energy potential is expected to increase slightly from winter to spring in northern Japan but decrease in the southern region. Additionally, the production of wind energy is anticipated to decrease by around 5% in Japan from summer to autumn. | [147] |
| China | RCP2.6, RCP4.5, RCP6.0 and RCP8.5 | 1971–2005 | 2066–2100 | There is no correlation in the projected mean wind speed for future periods compared to the past for a given AOGCM, despite interannual variability observed between 1971 and 2005. The projected interannual variation from 2066 to 2100 shows a minor dependence on the interannual variation from 1971 to 2005. | [148] |
| India | RCP 4.5 and RCP 8.5 | 1979–2005 | 2006–2032 | In all three locations, the average offshore wind capacity will measurably increase each year. | [149] |
| Caspian Sea | RCP4.5 and RCP8.5 | 1981–2000 | 2081–2100 | There is a forecast of a minor decrease in the annual wind energy production in the future. | [150] |
| Asia | SSP2-RCP2.6 and SSP2-RCP 6.0 | 1970–2000 | 2070–2100 | Wind energy in South East Asia increased by approximately 10%, as shown by statistically significant signals. Japan experienced a significant decrease in wind energy generation, whereas Korea's decrease was not statistically significant. | [9] |
| Middle East | RCP4.5 and SSP2–4.5, RCP8.5 and SSP5–8.5 | 1965–2005 | 2020–2059 and 2060–2099 | The eastward wind speed is expected to vary between −0.8 and 0.75 m/s, with Sudan and Mauritania experiencing the most significant increase and Algeria, Morocco, and western Libya witnessing the most significant decrease. Meanwhile, Saudi Arabia, Oman, and Yemen will experience the greatest alteration in the northward wind speed, whereas Egypt and eastern Libya will experience the least change. | [151] |
| Europe | | | | | |
| East Mediterranean | SRES A2 | 1961–1990 | 2071–2100 | Wind speed increased over land and decreased over the sea, except for a significant increase observed over the Aegean Sea. | [119] |
| Northern Europe | SRES A1B | 1961–1990 | 2020–2060 | The power potential has decreased by 2 to 6% in most areas. | [120] |
| Europe | RCP 8.5 | 1986–2005 | 2016–2035, 2046–2065 and 2081–2100 | There has been an increase in the Baltic Sea and a decrease in southern Europe. | [121] |
| Europe | RCP 4.5 and RCP 8.5 | 1979–2004 | 2021–2050 and 2061–2090 | Wind resources are expected to increase in North Africa and the Barents Sea, especially in the northern and western regions of the Black Sea area. | [122] |
| Greece | RCP 4.5 and RCP 8.5 | 2006–2015 | 2036–2045 | The average wind speed experienced a shift of about ±5%, which did not differ significantly between the different RCP scenarios. However, the variability of wind speed could reach up to ±20% on a monthly basis. | [124] |
| Mediterranean and the Black Sea | SRES A1B | 1961–1990 | 2021–2050 and 2061–2090 | The average wind speed and potential for wind power in the central Mediterranean Sea have decreased, except for in the Aegean Sea, Alboran Sea, and Gulf of Lion, where it has increased, showing a significant seasonality. | [125] |

**Table 3.** *Cont.*

| Study Area | Scenarios | Historical Period | Future Period | Projected Changes | Ref. |
|---|---|---|---|---|---|
| | | | **America** | | |
| United Kingdom | RCP 2.6, 6 and 8.5 | 1981–2000 | 2011–2030, 2041–2060 and 2071–2090 | The North Atlantic region and Northern Scotland have experienced the greatest increase in wind speed, whereas South England and the English Channel have recorded the largest decrease. Nevertheless, the prevailing model is inaccurate in reflecting the current distribution of wind resources in the UK. | [152] |
| Crotia | SRES A2 | 1961–1990 | 2011–2040 and 2041–2070 | Significant changes in average wind speed are expected along the coast and the neighboring mainland. By 2070, there is a potential increase of up to 50% in wind speeds during summers. | [128] |
| Italian peninsula | RCP 4.5 and RCP 8.5 | 1986–2005 | 2021–2050, 2051–2080, and 2071–2100 | The climate signal for the RCP 4.5 scenario is generally weak and statistically insignificant, whereas a more significant signal is observed in the medium and long term for the RCP 8.5 scenario. It is consistent with a decrease in wind production. In these regions, the RCP 8.5 scenario exhibits the least annual production decline, whereas the RCP 4.5 scenario presents moderate to long-term predictions of a slight upsurge in annual wind production, with a discernible upward trend mostly noticeable during spring. | [153] |
| Europe | RCP8.5 | 1976–2005 | 2011–2040, 2041–2070, and 2071–2100 | Overall, the situation is clearer with the projected increase in extreme winds across northern, central, and southern Europe, indicating more frequent occurrences. | [154] |
| Northern Europe | SSP585 | 1980–2009 | 2020–2049 | There is a general rise in the extreme winds over the North Sea and southern Baltic Sea, but a decrease over the Scandinavian Peninsula and most of the Baltic Sea. | [134] |
| Ireland | RCP 4.5 and 8.5 | 1981–2000 | 2041–2060 and 2081–2100 | Wind energy is projected to decrease by no more than 2% in the future. By mid-century, these changes will be more pronounced, especially for offshore areas under the RCP 8.5 scenario. In summer, the decrease in wind energy is expected to be less than 6%, whereas in winter it could increase by up to 1.1%. | |
| Central Europe | SSP2-RCP2.6 and SSP2-RCP 6.0 | 1970–2000 | 2070–2100 | There is an increase of about 10% in Central Europe (statistically significant signals). | [9] |
| Türkiye | SSP5–8.5 | 1980–2014 | 2080–2100 | The increases in wind speed in Aegean Sea, Aegean Region, Marmara Region, and Marmara Sea in summer are 5.9, 7.2,10, 14.7%, respectively, and the decrease in the winter season in Turkey is 4.1%. | [155] |

Climate forcing applied in the projections is described using the Representative Concentration Pathway (RCP) for CMIP5 generation, global climate model (GCM) simulations, or Special Report on Emission Scenarios (SRES) for CMIP3; CONUS, contiguous USA; Shared Socio-economic Pathways (SSP) for CMIP6 generation.

## 4. Discussion

As nations work towards transitioning to cleaner energy sources, wind power stands out as a significant contributor. Its potential to transform energy production has been accompanied by environmentally friendly characteristics, declining costs, and technological advancements, positioning it as a crucial element in the global shift towards cleaner energy portfolios. It is worth noting, however, that objective evaluations are necessary to determine the true impact of wind power. Furthermore, the advancement in energy storage technologies has bolstered the growth of wind energy toward sustainability. It facilitates the resolution of intermittent problems and guarantees reliable power supply.

This addresses a significant challenge in renewable energy. This interaction between wind energy and energy storage technologies not only ensures reliable power generation but also paves the way for exploring hybrid energy systems [48]. These hybrid systems, when synergistically integrated with complementary sources such as solar or energy storage systems, enhance the reliability and overall energy yield of renewable systems. Moreover, the rise of hydrogen-based economy proposes a new method of utilizing wind power, as electrolyzers powered by wind turbines can produce green hydrogen. This fuel source is viewed as a clean energy alternative for the future and has the potential to resolve the unpredictability challenges that accompany wind power. Furthermore, it can aid in establishing a more durable and sustainable energy infrastructure.

The COVID-19 pandemic has highlighted the resilience of renewable energy sources, particularly wind energy, in the face of economic and societal disruptions. The pandemic has led to a significant increase in wind energy installations, emphasizing the industry's potential to promote economic recovery and advance the transition toward clean energy [156]. These advancements reaffirm the adaptable nature of the wind energy industry and its interconnectedness with the wider environmental, economic, and political landscapes.

Climate change presents a challenge to wind resources by affecting large-scale atmospheric patterns that require precise climate models for full comprehension. The alterations within these patterns offer insights into how climate change may impact wind resources. Although there is hope that current and upcoming climate models can capture these changes, evaluating the impact of climate non-stationarity on wind resources poses difficulty due to discrepancies in scales as compared to current global circulation models [157]. The geographic dispersion of wind resources in mid-latitude areas, near major storm tracks, offers a unique opportunity for wind power integration [158,159]. Studying wind resources in these areas has significant implications for both wind power integration and climate change research. Understanding the geographical distribution of wind resources is vital for optimizing wind energy distribution across a grid, as maintaining a strong grid is essential for balancing electricity load and wind supply.

Accurate wind speed data are essential for designing and operating wind turbines efficiently. Despite its criticality, shortages of operational altitude wind speed data and inconsistent reanalysis data have impeded the evaluation of the reliability of model-based wind resource and operating condition projections [160]. Overcoming this challenge necessitates meticulous research into climate sensitivity, frequently inhibited by the computational costs associated with modeling phenomena at various scales. Identifying climate change signals, especially in extreme wind phenomena, is pivotal, yet challenging, due to model limitations [161], uncertainties in estimating extreme values [162], and the high temporal variability of rare events [147]. Furthermore, these signals rely on specific datasets, methodologies, and data periods. Several strategies and ongoing research efforts are being pursued to address these challenges, including the development of high-resolution regional climate models [163] and the integration of machine learning into climate modeling and climate change studies [164,165]. While researchers continue to explore the use of big data analytics and advanced simulation techniques, including high-resolution regional climate models, further studies are required to understand the correlations between climate patterns and wind energy production. Additional research is necessary to bridge gaps between climate modeling and energy systems and enable efficient energy planning for the future [166]. The varying climate projections across different regions highlight the intricate relationship between climate nonstationarity and wind power generation [157]. Robust and adaptable strategies are necessary to address the uncertainty surrounding the impacts of climate change in the design, operation, and formulation of wind energy policies, ensuring the reliability and sustainability of wind power in a changing climate.

In recent years, innovations have greatly increased the efficiency, reliability, and overall effectiveness of modern wind power systems [167]. Integration of digital technologies and intelligent solutions has led to a fundamental change in the management and optimization of wind power systems. Technological advances heavily focus on grid integration, which is

made possible by technological innovation streamlining the incorporation of intermittent energy sources, like wind energy, into current power grids. The industry's shift toward longer project lifetimes and the increasing adoption of repowering emphasizes the need to evaluate medium-term resource stability [167,168]. This focus on medium-term stability aligns with the expanding domain of offshore wind energy, stimulated by advances in offshore turbine design, foundational frameworks, and marine logistics [169]. Expanding into offshore areas not only benefits from the abundance of offshore wind resources but also addresses the complexities of land use while increasing the range of renewable energy scalability. As we investigate the microscale impact of climate phenomena on wind power generation, it is apparent that the structure and functioning of wind power facilities are interlinked with wind patterns and their connected parameters [157]. These parameters have a decisive influence on the level of fatigue and extreme mechanical loads [170]. Therefore, they play a significant role in determining the suitable classification of wind turbine for any given location [157].

Collaborative efforts in engineering, materials science, meteorology, and policy-making underscore wind energy's fundamental role in the global renewable energy landscape. Collaborating with experts from various fields demonstrates the significance of joint progression. By combining engineering expertise, a deep understanding of material science, meteorological knowledge, and strategic policy formulation, effective solutions have been created to address complex technical, environmental, and regulatory challenges.

## 5. Conclusions

In conclusion, this paper emphasizes the significant role of wind energy in combating global warming and promoting sustainable energy systems. It includes a comprehensive analysis that highlights the environmental, economic, and social advantages of wind energy, reaffirming its critical position in transitioning to cleaner energy portfolios. Additionally, our examination of advancements in wind energy, specifically in energy storage technologies and hybrid energy systems, illustrates the potential for improving the reliability and overall energy production of renewable systems. The regional availability of wind resources, particularly in areas with high wind abundance, bears important consequences for power integration and climate research.

Addressing the complexities of climate change research, especially the nonstationarity of climate effects on wind resources, necessitates innovative solutions and multidisciplinary collaboration. Obtaining precise wind speed data at higher altitudes remains crucial for efficient operation and accurate design, especially since climate models project varying impacts on wind resources throughout distinct regions. By examining the methodologies used in these studies, we gained insight into the reliability and robustness of the results. This underscores the importance of using credible models and scenarios in climate change impact assessments. The impact of climate change on wind energy potential is not uniform. While some regions, such as North America, may experience increased wind speeds, others, such as the Mediterranean, may face potential reductions. Of particular note is the forecast for a potential long-term increase in wind speeds in Northern Europe. The interplay between climate non-stationarity and wind power generation is complex, leading to a range of projections. While there is consensus that climate change will affect wind speeds and energy production, the details, including location and magnitude, remain uncertain. These findings have important implications for the wind energy sector. They underscore the importance of region-specific assessments and adaptation strategies. In addition, the uncertainty surrounding the exact nature of climate change impacts underscores the need for continued research, innovation, and policy development to ensure the reliability and sustainability of wind energy in a changing climate. Therefore, region-specific strategies are crucial for optimizing the distribution of wind energy and harmonizing electricity markets. At a micro scale, this examination has delved into the complex correlations between wind patterns, their relevant parameters, and their impacts on the engineering and operational



aspects of wind farms. This comprehensive approach informs decisions regarding wind turbine classification, which directly affects project viability and operational efficiency.

Innovation in technology has played a vital role in promoting the performance, dependability, and productivity of modern wind energy systems. Incorporation of digital technologies and intelligent solutions has drastically transformed the management and optimization of wind energy systems. The industry's shift towards longer project lifetimes and re-powering underscores the significance of assessing the stability of resources over the medium term. Unlocking the full potential of wind energy requires a thorough comprehension of meteorological conditions and a steady commitment to technological advancement. Wind power possesses environmentally friendly attributes and decreasing costs, making it ideally situated to play a pivotal role in a more sustainable worldwide energy infrastructure.

**Author Contributions:** Conceptualization, T.K. and A.D.Ş.; methodology, writing—original draft, preparation, T.K.; writing—review and editing, T.K. and A.D.Ş. All authors have read and agreed to the published version of the manuscript.

**Funding:** This research did not receive external funding.

**Institutional Review Board Statement:** Not applicable.

**Informed Consent Statement:** Not applicable.

**Data Availability Statement:** In this study, no new data were created or analyzed in this study. Data sharing is not applicable to this article.

**Conflicts of Interest:** The authors declare no conflict of interest.

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
