# Peer review of "Implications of Climate Change on Wind Energy Potential"

_sustainability, doi:10.3390/su152014822_

Round 1

Reviewer 1 Report

- Specific result holud be added at the end of the abstract and Ch. 5; conclusion.

-All abrivations should be explained at the first place.

- References: No3, year was missed, 

- Ref No. 51, year was missed, 

- Total number of references are too much. 

- No any papers were referred written by national authors, 

Reviewer 2 Report

This is a review-type article. 

I have no further comments on this. 

Author Response

Thank you very much for taking the time to review this manuscript. Although you have no further comments on this study, we have updated and enhanced our study in light of feedback from valuable reviewers.  

Reviewer 3 Report

This manuscript presents an overview of the role of wind energy in addressing global warming and advancing toward sustainable energy systems. The authors emphasize the environmental, economic, and social benefits of wind energy, and discuss the challenges posed by uncertainties in wind modelling and the impact of climate change on wind patterns.   The content is indeed of interest to the scientific community; however, I believe there is a great potential for the improvement of this manuscript.  

I really would like to see what new opinions I can learn based on this manuscript, instead of just a collection of literatures.  Hence, I would kindly suggest that the authors could improve the discussion section by highlighting their own opinions and analysis.  What are the major gains after reviewing so many literatures?  A good literature review should offer extra values to the existing knowledge.  

Another related suggestion is to improve writing by merging some paragraphs together. There are lots of paragraphs only containing two or three sentences.  This writing style can easily distract readers and the readers cannot follow the key idea that you would like to express.

English language needs to be improved.

Round 2

Reviewer 3 Report

I am satisfied with the revision.  I thank the authors taking efforts to incorporate my comments to the manuscript and improve it.  I think the current version is suitable for publication.